# Role of Platinum Nanozymes in the Oxidative Stress Response of *Salmonella* Typhimurium

**DOI:** 10.3390/antiox12051029

**Published:** 2023-04-29

**Authors:** Mireya Viviana Belloso Daza, Anna Scarsi, Francesca Gatto, Gabriele Rocchetti, Pier Paolo Pompa, Pier Sandro Cocconcelli

**Affiliations:** 1Department for Sustainable Food Process (DISTAS), Università Cattolica del Sacro Cuore, 29122 Piacenza, Italy; 2Nanobiointeractions & Nanodiagnostics, Istituto Italiano di Tecnologia (IIT), 16163 Genova, Italy; 3Department of Chemistry and Industrial Chemistry, University of Genoa, Via Dodecaneso, 16146 Genova, Italy; 4Department of Animal Science, Food and Nutrition (DiANA), Università Cattolica del Sacro Cuore, 29122 Piacenza, Italy

**Keywords:** platinum nanozymes, reactive oxygen species, oxidative stress, *Salmonella*, antimicrobial nanoparticles

## Abstract

Platinum nanoparticles (PtNPs) are being intensively explored as efficient nanozymes due to their biocompatibility coupled with excellent catalytic activities, which make them potential candidates as antimicrobial agents. Their antibacterial efficacy and the precise mechanism of action are, however, still unclear. In this framework, we investigated the oxidative stress response of *Salmonella enterica* serovar Typhimurium cells when exposed to 5 nm citrate coated PtNPs. Notably, by performing a systematic investigation that combines the use of a knock-out mutant strain 12023 HpxF^-^ with impaired response to ROS (Δ*katE* Δ*katG* Δ*katN* Δ*ahpCF* Δ*tsaA*) and its respective wild-type strain, growth experiments in both aerobic and anaerobic conditions, and untargeted metabolomic profiling, we were able to disclose the involved antibacterial mechanisms. Interestingly, PtNPs exerted their biocidal effect mainly through their oxidase-like properties, though with limited antibacterial activity on the wild-type strain at high particle concentrations and significantly stronger action on the mutant strain, especially in aerobic conditions. The untargeted metabolomic analyses of oxidative stress markers revealed that 12023 HpxF^-^ was not able to cope with PtNPs-based oxidative stress as efficiently as the parental strain. The observed oxidase-induced effects comprise bacterial membrane damage as well as lipid, glutathione and DNA oxidation. On the other hand, in the presence of exogenous bactericidal agents such as hydrogen peroxide, PtNPs display a protective ROS scavenging action, due to their efficient peroxidase mimicking activity. This mechanistic study can contribute to clarifying the mechanisms of PtNPs and their potential applications as antimicrobial agents.

## 1. Introduction

The constant increase of bacterial resistance to antibiotics represents one of the greatest worldwide health challenges in the recent years, posing the necessity to find new longer-term solutions for successful control of bacterial infections. In this regard, there is a growing interest in developing and exploiting engineered nanomaterials that could integrate novel antibacterial functionalities [1]. Among the currently available nanotechnology tools, metallic and metallic oxide nanostructures have been proposed as potential candidates to overcome the drawbacks of antibiotics due to their peculiar chemical properties, high surface-to-volume ratio and their potential antibacterial activity [2,3]. Several nanoparticles (NPs), such as Ag, Pd, Au, Cu, ZnO and TiO_2_, have shown promising results [4], even if concerns about their cytotoxicity limit their practical use [5].

The mechanisms of antimicrobial action of nanoparticles depend on the type of microorganism and the physicochemical characteristics of the nanoparticles, such as their size. In the last decades, great interest has increased in food safety and biomedical research of metallic nanoparticles, especially Ag and Au. These can reduce oxidative stress through their efficient radical scavenging activities. Additionally, silver nanoparticles (AgNPs) possess intrinsic broad-spectrum antimicrobial characteristics [6,7,8,9] and have been widely adopted as effective bactericidal agents [10,11,12,13]. AgNPs also confer a broad range of antimicrobial activity and are currently used to control bacterial infections, contributing significantly to the field of biomedicine and with great potential for application in the food industry [14]. The antimicrobial activity of AgNPs in *Salmonella* has been attributed to the accumulation of reactive oxygen species (ROS) and release of silver ion, affecting so the permeability of the inner membrane and causing membrane dysfunction leading to aggregation and clumping of cytoplasm structures [15,16]. Nevertheless, the increasing use of silver in a great number of commercial and medical tools is leading to the development of bacterial molecular strategies of resistance. Mechanisms by which bacteria become resistant to silver involve the reduction of Ag^+^ to its less toxic neutral oxidation state, or the employment of active efflux from cell. Moreover, it has been recently reported that, after repeated long-term exposure to subinhibitory concentrations of AgNPs, Gram-negative bacteria can promote the aggregation of silver NPs by the production of the bacterial flagellum protein flagellin, thus evading the antibacterial effect [17]. Furthermore, AgNPs present an important threat due to their toxicity to human cells [18,19]. Therefore, it is essential to exploit new antimicrobial nanomaterials with a proper biocompatibility. Gold, platinum and palladium-based NPs have been revealed as safer antimicrobial candidates with no acute toxicity reported [20,21]. Notably, together with the enzyme mimetics of PtNPs, they have presented insignificant cytotoxic effects with cells and cell-compatible materials, being great candidates in cancer therapy [22,23].

Bacterial inhibition through metal nanoparticles has been attributed to mechanisms such as metal ions disrupting enzymes, DNA, cellular transport and cell membrane [24]. Additionally, evidence suggest that the antibacterial properties of noble metal-based NPs are usually attributed to their oxidase- and peroxidase-like activity. In particular, reactive oxygen species (ROS) are able to oxidize diverse cellular components [25]. As a promising alternative to natural enzymes, nanozymes are catalytic nanomaterials possessing several advantages, such as low-cost synthesis, room temperature stability and the possibility to be employed in harsh conditions maintaining high efficiency [26]. Furthermore, the catalytic activity and thus the enzyme-like behaviour of NPs could be controlled tuning several features such as size, shape and exposed facets. For example, shape and facet-dependent antibacterial activities of Pd nanocrystals have been recently reported, exhibiting high oxidase-like properties [27]. This catalytic activity efficiently inhibited the proliferation of both Gram-positive and Gram-negative bacteria, albeit at high NP concentrations. This behaviour could be attributed to the different membrane-penetration capacity of Pd nanocrystals, even though the precise mechanisms involved need further clarification.

In this framework, the aim of our investigation was to perform a mechanistic study to understand the dynamics of the enzyme-like activity of 5 nm PtNPs using a model of *Salmonella enterica* Typhimurium. This Gram-negative pathogen is responsible for gastroenteritis and forms biofilms that enhance its dissemination throughout the environment [28]. Particularly, *Salmonella* has developed a series of antioxidant defences to scavenge and reduce ROS [27]. Notably, KatE, KatG, KatN catalases and AhpCF (alkyl hydroperoxide reductase) and TsaA (thiol-specific antioxidant) reductases have been shown to play an important role in H_2_O_2_ scavenging and form a redundant antioxidant arsenal essential for survival and replication within host cells, strongly associated with increased virulence [29,30,31]. In light of this, we selected the knock-out mutant HpxF^-^ (Δ*katE* Δ*katG* Δ*katN* Δ*ahpCF* Δ*tsaA*), with impaired response to ROS, and its parental wild-type strain in order to investigate the oxidative stress response of *Salmonella* Typhimurium cells when exposed to PtNPs.

## 2. Materials and Methods

### 2.1. Bacterial Strains and Culture Conditions

*Salmonella enterica* serovar Typhimurium 12023 and its knock-out mutant HpxF^-^ (Δ*katE* Δ*katG* Δ*katN* Δ*ahpCF* Δ*tsaA*) were cultured as described by He’bard et al., 2009. Briefly, parental strain 12023 was grown in BHI (Oxoid) and 12023 HpxF^-^ was grown in BHI supplemented with 25 mg/L kanamycin (Sigma-Aldrich, Germany) and incubated at 37 °C.

### 2.2. Platinum Nanoparticle Synthesis, Functionalization and Characterization

In this study, 5 nm spherical citrate-coated platinum nanoparticles were synthesized by wet chemical reduction, following a previously reported protocol with some optimizations [32].

Briefly, 320 µL of a 0.5 M solution of chloroplatinic acid hexaydrate (Sigma-Aldrich) was added to 160 mL of MilliQ water at room temperature, followed by a trisodium citrate solution (Sigma-Aldrich) and a sodium borohydride solution (Sigma-Aldrich) as capping and reducing agents, respectively. The reaction mixture was then heated up to 75 °C and left under stirring for 30 min. The amounts of chloroplatinic acid hexaydrate and reducing agents were finely tuned in order to obtain the desired size of PtNPs.

PtNP monodispersity was analysed by Transmission Electron Microscopy (TEM) and Dynamic Light Scattering (DLS) (Figure 1). Since we observed extensive aggregation of PtNPs when dissolved in BHI medium, PtNP stability was improved by functionalization with Bovine Serum Albumin (BSA) (Sigma-Aldrich) [33,34]. BSA-coating was performed through physical adsorption of the protein at the surface of the nanoparticles. First, the pH of the aqueous solution of 5 nm PtNPs was raised to 7.5, and then the BSA solution was added. The reaction flask was kept under stirring for 30 min at room temperature, and then the solution was washed using 30 K Amicon Ultra centrifugal filters. The final dose of PtNPs was determined by ICP-OES analysis. For PtNPs characterization, TEM analysis was performed by using a JEOL JE-1011 microscope with thermionic source (W filament). Accelerating voltages: 100 kV; conventional TEM imaging: bright field; TEM resolution = 4.0 Å (100 kV). For DLS analysis, PtNPs and PtNPs-BSA suspensions were diluted in MilliQ water and BHI medium up to optimal dose, and the spectra were recorded at room temperature by Zetasizer Nano Range (Malvern-PANalytical, United Kingdom) as frequency distribution of intensity.

### 2.3. Biocidal Effect of PtNPs

For the determination of the bacterial growth inhibition, different doses of platinum nanoparticles (0, 5, 10, 20, 50 and 100 mg/L) were prepared in 10 mL of BHI. Next, 100 µL of log phase *Salmonella* Typhimurium 12023 and 12023 HpxF^-^ were inoculated to each tube containing the different PtNPs doses and grown at 37 °C overnight. Incubation was performed under aerobic and anaerobic conditions. To ensure purgation of micromolar concentrations of O_2_ in the BHI medium, saline solution and PtNPs solution, they were submitted to nitrogen bubbling for 30 min under aseptic conditions. All tests conducted under anaerobiosis were then prepared, performed and evaluated in an anaerobic sterile chamber. After incubation, the cultures were serially diluted and plated on BHI, with respective antibiotics when necessary. Plate counts were performed in triplicate.

The biocidal effect was performed in aerobiosis and was determined as follows: log phase cultures of parental and knock-out strains were treated PtNPs at doses of 20 and 50 mg/L. After 1 h of incubation, PtNPs were removed by filtering the suspension through a 0.22 µm cellulose filter (Merck MF-Millipore, USA). Filters were washed with sterile saline solution and then resuspended in initial volume of BHI, serially diluted and plated onto respective agar plates. A negative control was included.

### 2.4. Hydrogen Peroxide Scavenging and Sensitivity

To test the scavenging capacity and potential sensitivity to H_2_O_2_, *Salmonella* strains were subjected to H_2_O_2_ in the presence of PtNPs. Incubation was performed in anaerobiosis as described above. Briefly, overnight cultures of 12023 and 12023 HpxF were diluted to an OD of 0.1 and mixed with 10 µg/mL PtNPs. Next, different concentrations of H_2_O_2_ were added to the solution (0, 0.001, 0.01, 0.1, 0.5, 1, 2, 5 and 10 mM). A control with 0 µg/mL PtNPs was included. Cells were incubated at 37 °C and optical density was measured at 2, 4, 8 and 20 h. All samples were prepared in triplicate.

### 2.5. Untargeted Metabolomics by UHPLC-HRMS

Metabolomic profiling of the oxidative stress of *Salmonella* Typhimurium induced by PtNPs was determined using high-resolution mass spectrometry HRMS) performed on a Q-Exactive™ Focus Hybrid Quadrupole-Orbitrap Mass Spectrometer (Thermo Scientific, Waltham, MA, USA) coupled with a Vanquish ultra-high-pressure liquid chromatography (UHPLC) pump and equipped with heated electrospray ionization (HESI)-II probe (Thermo Scientific, USA).

Parental and mutant strains were incubated overnight with PtNPs doses of 0, 20 and 50 mg/L. Next, cell cultures were centrifuged for 10 min at 10,000× *g*. Both pellet and supernatant were kept for analysis. For the extraction step, pellet and supernatant were treated using a 1:20 ratio with the extraction buffer, consisting of 80% methanol (Carlo Erba) acidified with 0.1% formic acid (Carlo-Erba). Thereafter, they were incubated for 10 min at maximum power using an ultrasound assisted extraction step. The samples were centrifuged at 4 °C for 10 min at 10,000× *g*. For subsequent instrumental analysis, extracted pellet samples were added to UHPLC vials and supernatant samples were filtered using 0.22-micron syringe-filters before adding them to UHPLC vials. The chromatographic separation was achieved by using a water-acetonitrile (both LC-MS grade, from Sigma-Aldrich, Milan, Italy) gradient elution (6–94% acetonitrile, duration time: 35 min) using 0.1% formic acid as phase modifier. For the separation step, the Agilent Zorbax Eclipse Plus C18 column (50 × 2.1 mm, 1.8 μm) was selected. The HRMS conditions were adapted as described previously [35]. The flow rate was 200 μL/min, and full scan MS analysis was chosen in the range 100–1200 *m/z*, with a positive ionization mode and a mass resolution of 70,000 at *m/z* 200. The injection volume was 6 μL. The parameters related to Ion Trap, such as the automatic gain control target (AGC target) and the maximum injection time (IT), were previously optimized [35]. Furthermore, pooled quality control (QC) samples were randomly acquired in a data-dependent (Top N = 3) MS/MS mode, using a full scan mass resolution of 17,500 at *m/z* 200. The Top N ions were selected for fragmentation under typical Normalized Collisional Energy values (10, 20, 40 eV). The HESI parameters for both MS and MS/MS are reported elsewhere [35]. The mass spectrometer was calibrated using Pierce™ positive ion calibration solution (Thermo Fisher Scientific, San Jose, CA, USA) before the analysis. The raw data (.RAW files) were then further processed using the software MS-DIAL (version 4.70) [36]. In particular, automatic peak finding, LOWESS normalization and annotation via spectral matching (against the database MoNA—Mass Bank of North America) were carried out. The mass range 100–1200 *m/z* was searched for features with a minimum peak height of 10,000 cp*s.* The MS and MS/MS tolerance for peak centroiding was set to 0.01 and 0.05 Da, respectively, excluding the retention time information from the calculation of the total score. Accurate mass tolerance for identification was 0.01 Da for MS and 0.05 Da for MS/MS. and the identification step was based on mass accuracy, isotopic pattern and spectral matching. The total identification score cut-off was 60%, considering the most common HESI + adducts, while the gap filling using peak finder algorithm was used to fill in missing peaks, considering 5 ppm tolerance for *m/z* values. The software MS-Finder [37] was then used for in silico fragmentation of not fully annotated mass features, using Lipid Maps and FoodDB libraries. Therefore, under our experimental conditions, a level 2 of confidence in annotation was achieved [38].

### 2.6. Malondialdehyde TBARS Assay

The ratio between GSH and oxidized glutathione (GSSG) clarifies on how 12023 and HpxF^-^ would react to ROS in terms of cell membrane oxidation in the presence of sublethal PtNPs doses. Lipid oxidation was determined by the TBARS assay as described previously [39,40]. Parental and mutant strains were incubated overnight with PtNPs doses of 0, 20 and 50 mg/L. After, cell cultures were centrifuged for 10 min at 10,000× *g*. For the extraction step, both the pellet and supernatant were treated with a 1:2 ratio of extraction buffer that consists of 0.1% (*w/v*) Trichloroacetic acid (TCA) in dH_2_O. Samples were then incubated under ultrasound at maximum power for 10 min. Followed by a centrifugation step (10,000 rpm for 10 min), both the supernatant and pellet were added to one solution containing 20 % of TCA (*w/v*) and 0.65% of TBA (Thiobarbituric acid) (*w/v*) and to a second solution containing 20% of TCA. The samples were then mixed by inversion, incubated for 15 min at 95 °C and cooled down to stop the reaction. After centrifugation for 10 min at 10,000 × g, the samples were analysed using a spectrophotometer at an optical density of 532 nm. For MDA determination, a molar extinction coefficient of 155 cm^−1^ mM^−1^ was used. The results were finally expressed as nM MDA equivalents (n = 3).

### 2.7. Statistical Analysis

ANOVA analysis, with subsequent Tukey’s significant difference test (*p*-value of 0.05), was used to compare the data obtained from each experiment using IBM SPSS Statistics (Version 25; IBM, Armonk, NY, USA). All the data obtained in triplicate were reported as mean values ± standard deviation (SD). ANOVA results for the MIC determination, the biocidal effect, and the combined effect of H_2_O_2_ for both strains, are listed in Appendix A.

The metabolomics dataset was exported into the software SIMCA 13 (Umetrics, Malmo, Sweden). After data normalization by median, Log transformation (base 10) and Pareto scaling, a supervised multivariate statistical analysis was carried out. In more detail, the data analysis was based on supervised orthogonal projections to latent structures discriminant analysis (OPLS-DA) to provide a tentative discrimination of the samples according to fixed observation groups, thus maximizing their differences. From a statistical point of view, the OPLS-DA considers only the Y-predictive variation, thus eliminating that not directly correlated with Y in the data matrix (i.e., metabolomics dataset). Additionally, the OPLS-DA model validation parameters (goodness-of-fit R^2^Y together with goodness-of-prediction Q^2^Y) were inspected, considering a Q^2^Y prediction ability of >0.5 as the acceptability threshold. The OPLSA prediction model was built considering the metabolomic profile of treated and untreated supernatant and cells of parental strain 12023 and mutant strain 12023 HpxF-.

## 3. Results and Discussion

### 3.1. Synthesis and Characterization of Platinum Nanoparticles

The synthesized 5 nm PtNPs were monodispersed and homogeneous in shape and size, as illustrated in the TEM image (Figure 1A) and corresponding size distribution (Figure 1B). Moreover, they were stable in water and showed a hydrodynamic radius around 8–9 nm (Figure 1C, red curve). However, when added to BHI medium, PtNPs showed a rapid and significant aggregation, causing a strong peak shift in the DLS spectrum (Figure 1C, blue curve). To increase their stability in this medium, PtNP surface was coated with BSA, through physical adsorption of the protein in aqueous solution. As illustrated in Figure 1D, the presence of the adsorbed protein led to a peak shift of about 20 nm in water (red curve) and a similar peak is shown in BHI medium (blue curve), confirming that BSA coating ensured a relatively high stability of PtNPs.

### 3.2. Effect of PtNPs on Salmonella Typhimurium Growth

The first step of our assessment involved the evaluation of the bacterial growth inhibition capacity of PtNPs on *Salmonella* Typhimurium 12023 and its derivative mutant 12023 HpxF^-^, to investigate if the lack of ROS-coping enzymes may result in different cell viability. Figure 2 shows that the growth of 12023 was only partially limited up to 20 mg/L particle concentration. A slightly more effective reduction is observed at 50 and 100 mg/L with 0.53 and 0.64 Log CFU reduction, respectively. No significant differences were detected between growth under aerobic or anaerobic conditions. On the contrary, a statistically significant reduction (*p* < 0.05) in cell numbers was observed in 12023 HpxF^-^ as a function of particle concentration. The mutant strain seemed to be more sensitive to the PtNPs in aerobic conditions with respect to anaerobiosis, particularly at a dose of 5 mg/L.

In a second experiment, we analysed the PtNPs effect on cells exposed for 1 h to sublethal doses of PtNPs (20 and 50 mg/L) (Figure 3). After 1 h of exposure at 20 mg/L, cell viability was reduced of 0.46 and 0.45 Log CFU for parental and mutant strains, respectively. At 50 mg/L PtNPs, a statistically difference was observed between the parental strain 12023 and mutant 12023 HpxF^-^, with a 1.24 Log CFU reduction.

It has been reported that PtNPs have outstanding catalytic activity and exhibit typical kinetics of oxidases [41,42]. Interestingly, Song and colleagues reported that PtNPs coupled on deposited Multiwalled Carbon Nanotubes presented oxidase activity generating superoxide O_2_^•−^, from dissolved O_2_ [43]. In fact, this catalytic activity mimics one of the best-characterized sources of ROS during host cell–pathogen interactions, namely the NADPH oxidase [44]. The accumulation of superoxide causes bacterial membrane lipid peroxidation, which increases the cell permeability causing the uncontrolled transport of intra- and extracellular molecules, finally leading to cell death [45]. Our data indicate that the strain 12023 HpxF^-^, with impaired response to oxidative stress, is significantly more susceptible than its parental strain. The combined effect of *Salmonella* Typhimurium ROS defence enzymes such as alkyl hydroperoxide reductases AhpCF, TsaA and catalases KatE, KatG and KatN enables possibility of two types of scavenging systems that contribute to oxidative stress survival. Hence, while catalases act as the first line of oxidative stress defence by scavenging H_2_O_2_, alkyl hydroperoxide reductases eliminate micromolar concentrations of H_2_O_2_ and other hydroperoxides [29]. These reductases are part of peroxiredoxins, which reduce organic hydroperoxides to alcohols and hydrogen peroxide to water at the expense of NADH or NADPH [46]. Additionally, Hébard and colleagues (2009) showed that *S.* Typhimurium Kat^-^ mutants (Δ*katE* Δ*katG* Δ*katN*) and reductase Ahp^-^ mutants (Δ*ahpCF* Δ*tsaA*) were still able to scavenge H_2_O_2_ due to the compensatory regulation of the other enzymes [31]. Previous studies have shown similar results with *Salmonella enterica* ser. Infantis exposed to 50 mg/L of PtNPs had a reduction of 1 Log CFU [47].

The antibacterial activity of PtNPs has been demonstrated in several studies [23]. Hashimoto and colleagues demonstrated a MIC of 400 mg/L with an NP size of 5 nm against *S. aureus* [48]. Chlumsky et al. indicated strong inhibitory effect of bacterial growth at a dose of 50.5 mg/L, showing inhibitor values between 84–99% against *S. aureus*, *L. monocytogenes*, *E. coli* and *Salmonella* Infantis [21]. Nevertheless, minimum bactericidal and bacteriostatic doses of PtNPs reported in literature are difficult to compare not only due to the synthesis methodology of NPs but also because MIC values are often expressed as inhibition zones (mm) [49], micromolar doses [50] or optical density values [21]. Additionally, the synthesis of PtNPs diverges across studies, rendering it more difficult to perform a comparison assessment [51].

### 3.3. The Combined Effect of PtNPs and H_2_O_2_ on Salmonella Typhimurium

After analysing the inhibitory effect of PtNPs-induced oxidative stress on *Salmonella* Typhimurium, we tested whether the presence of exogenous H_2_O_2_ would cause the PtNPs to act as ROS scavengers and protect the cell, or as ROS enhancers by boosting the combined oxidative stress effect. To this end, we exposed cells to increasing concentrations (0.001–10 mM) of H_2_O_2_ in the presence or absence of the sub-inhibitory dose of 10 µg/mL PtNPs (Figure 4). As shown in Figure 4A, the growth of strain 12023 was not affected by H_2_O_2_ equal or below 1 mM, whereas at higher concentrations (2, 5, 10 mM) of H_2_O_2_, the growth was significantly inhibited. Interestingly, when 10 µg/mL of PtNPs were added, a protection effect was observed, with 12023 being able to grow with 2, 5 and 10 mM of H_2_O_2._ Conversely, the mutant 12023 HpxF^-^ presented substantially higher susceptibility to hydrogen peroxide, with growth inhibition at H_2_O_2_ concentrations higher than 0.001 mM. These data are consistent with a previous study [31] which showed that 12023 HpxF^-^ accumulates H_2_O_2_ during aerobic growth much faster than its parental strain, and when exposed to exogenous hydrogen peroxide, it was able to survive at a concentration of < 0.001 mM H_2_O_2_. Notably, the presence of 10 µg/mL PtNPs in the growth medium exerted a protection effect enabling the mutant strain to grow at a 10-fold-higher concentration of H_2_O_2_ (0.01 mM, Figure 4D). This indicates that the addition of low doses of H_2_O_2,_ does not increase oxidative stress, but protects the cells by shifting the catalytic ability of the nanozymes to scavenge exogenous ROS mimicking peroxidase activity, rather than generating them. In fact, other studies have shown that a PtNPs catalytic activity can be modulated depending on external conditions. In the presence of PtNPs, the H_2_O_2_ dissociates and generates OH^•^ on the NP surface. Since this species is not stable, the H_2_O_2_ was ultimately reduced in a peroxidase-like manner to H_2_O and O_2._ The protection capacity of PtNPs against oxidative stress has been observed in mammalian cells [52]. Particularly, lung cancer cells when exposed to 100 µg/mL PtNPs and 350 µM H_2_O_2_ and rat skeletal L6 cells when exposed to 10 µg/mL PtNPs and 10 μM H_2_O_2_ [53]. To the best of our knowledge, there are no studies that show the protection capacity of PtNPs in the presence H_2_O_2_ towards bacteria. This effect is more evident in the parental strain, being protected with the *Salmonella* innate ROS coping machinery, together with the peroxidase-like remotion of H_2_O_2_ by the nanoparticles. Furthermore, the results obtained with the mutant strain clarify that the peroxidase-mimicking activity of the PtNPs was enough to protect *katE katG katN ahpCF tsaA*-deficient cells by correcting the stress effects that preserve the bacterial growth. In a previous study, it was demonstrated that the failure of the induction of *ahpC*, *katG* and *katE* gene expression in *E. coli* resulted in higher susceptibility against H_2_O_2_-induced oxidative stress [54]. Likewise, in another report by Lui and colleagues, when exposing *Salmonella* Enteritidis to 3 mM of H_2_O_2_, genes *katG* and *ahpCF* boosted about 31–40 and 41–50 fold-change increase in differential expression [55].

### 3.4. PtNPs-Induced ROS, Affect Membrane Lipids and Oxidize DNA

Taking into consideration the previous results, untargeted metabolomics using UHPLC-HRMS was exploited to evaluate the effect of PtNP-induced oxidative stress on cell metabolism. In particular, we analysed the metabolomic profile of cell extracts and supernatants of both *Salmonella* Typhimurium 12023 and 12023 HpxF^-^ strains. As shown in Figure 5, OPLS-DA (Orthogonal Projections to Latent Structures Discriminant Analysis) scatter plot of extracts from supernatant and pelleted cells are reported. A dose-response effect on the metabolomic profiles of cells and supernatants was observed, as all samples of both strains exposed to the same doses of PtNPs clustered together and were distinct from other samples treated with different doses of PtNPs. No significant differences were observed in the metabolome profiles between the two studied strains by using this supervised approach.

In contrast, when the oxidative stress markers for Glutathione oxidation, DNA oxidation and lipid peroxidation were separately analysed, the two strains responded differently to oxidative stress (Figure 6). Remarkably, our results show that the exposure dose to 20 and 50 mg/L caused the GSH/GSSG ratio to decrease in the parental strain, suggesting that GSH oxidizes quicklier to combat oxidative stress effectively. On the contrary, the decrease of the ratio in 12023 HpxF^-^ is slower due to a decompensation of ROS-metabolism machinery, showing a less effective way of preventing oxidative stress caused by ROS. GSH oxidation induced by NPs (Ag) has been observed mainly in eukaryotic cells [56,57,58]. In the presence of GSH, peroxidase-catalysed one-electron oxidation leads to ROS formation with GSSG production. Therefore, decrease of GSH levels could be supported by the pro-oxidant property of the PtNPs as well as indicating disruption of the intracellular redox state [59]. Glutathione reduction systems have a key role in maintaining the reduced environment in *Enterobacteriaceae* and when ROS are generated, GSH is oxidized to GSSG, resulting in a decrease in GSH and increase in GSSG content in cellular extracts [37].

Furthermore, the DNA oxidation was analysed by measuring three different markers of oxidative damage, namely, 8-oxo-G, 8-oxo-2dG and 8-oxo-2dA [60]. The concentration of the three oxidative markers increased proportionally to the dose of PtNPs. However, this effect was higher in mutant strain 12023 HpxF^-^, as shown by the ratio between the concentration of oxidative markers in the mutant and parental strain, as reported in Figure 6. Moreover, the intracellular interactions of the PtNPs with bacterial DNA have been previously reported with *Salmonella* Enteritidis, and it was shown that PtNPs affected the DNA through ROS damage [61].

Stress-related lipids oxidation was qualitatively assessed by untargeted metabolomics with markers for cell envelope fatty acid compounds in *Salmonella,* as a function of the fold change of mutant over parental strain (Figure 6). Overall, at 50 mg/L PtNPs, the fold change is higher when compared to the lower PtNP doses. In addition to this, previous studies have stated that H_2_O_2_ up-regulates efflux pumps conferring an additional protection towards ROS [62]. Thus, the mutant strain lacking the catalases and reductases accumulates ROS in a more rapid manner downregulating the expression and the function of efflux pumps, and so causing terminal damage to the cell envelope. Other studies show that, under exogenous stress, *Salmonella* Typhimurium modifies its outer membrane fatty acid composition, increasing unsaturated oleic and linoleic acid and resulting in enhanced fluidity of the cellular membrane that affects permeabilization [63,64].

This is in accordance with our results when analysing the high degree of lipid peroxidation of the mutant strain in comparison with its parental strain (Figure 6). Moreover, the determination of MDA, the benchmark method for ROS-induce lipid-peroxidation [65] showed that in parental strain 12023, no MDA was detected at any dose of PtNPs, whereas in mutant strain 12023 HpxF^-^, gradually increasing values of 0.001, 0.002 and 0.004 nM of MDA were detected at 0, 20 and 50 mg/L, respectively.

Planchon et al. used a combined omics-based approach to evaluate the metabolic response of *E. coli* when exposed to TiO-NPs, where the upregulation of proteins involved in oxidative stress protection mechanisms was demonstrated [66]. To the best of our knowledge, microbial oxidative stress response induced by PtNPs has not been studied following a metabolomics approach, therefore, the data presented here aid in understanding the response of *Salmonella* and the role of ROS-depleting enzymes.

## 4. Conclusions

Platinum nanoparticles have awakened interest as antimicrobial agents due to their strong catalytic activities, mimicking oxidase and peroxidase enzymes. Herein, we studied the oxidative stress response of *Salmonella enterica* Typhimurium when exposed to PtNPs and their role in ROS-scavenging. In line with other studies, our results demonstrated that, at high doses, PtNPs exert antibacterial activity, principally due to their oxidase-like properties. A significantly stronger effect was observed in mutant strain 12023 HpxF^-^ (Δ*katE* Δ*katG* Δ*katN* Δ*ahpCF* Δ*tsaA*) compared with the limited activity on the parental strain, especially in aerobic conditions. Moreover, metabolomic analysis for measuring oxidative stress including markers for lipid, glutathione and DNA oxidation, confirmed the lower efficiency of 12023 HpxF^-^ to cope with PtNPs-based oxidative stress. Conversely, when combined with other ROS such as H_2_O_2_, PtNPs switched their catalytic activity mimicking peroxidases promoting ROS scavenging, thus protecting to a higher degree the parental strain from oxidative damage. This mechanistic study shed light on the understanding that the efficiency of *Salmonella* Typhimurium to tolerate oxidative stress induced by PtNPs is undertaken by complex enzymatic machinery activated to withstand ROS. Nonetheless, only at high doses of PtNPs was a significant reduction observed, indicating that PtNPs have limited action against *Salmonella* and a surprising protection effect when combined with H_2_O_2_. Future research should focus on the evaluation of PtNPs as antibacterial candidates alone or in combination with other bactericidal agents against microbial pathogens, especially *Salmonella*, with implications for both in clinical settings and the food safety field.

## Figures and Tables

**Figure 1 antioxidants-12-01029-f001:**
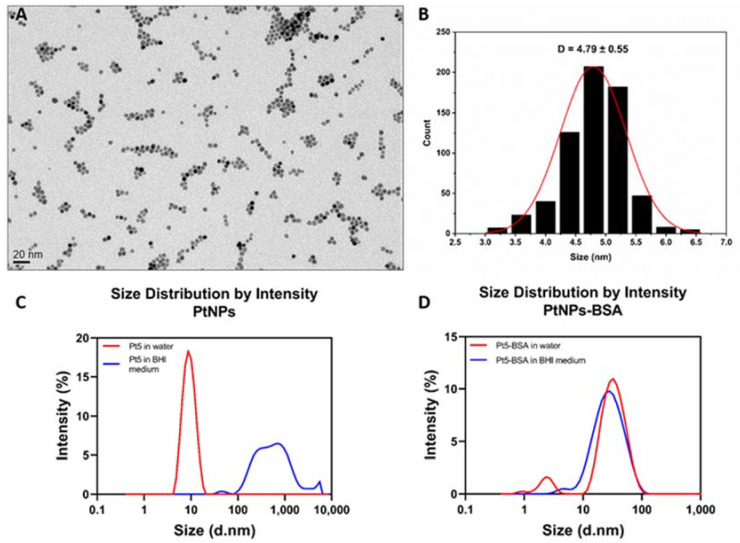
Characterization of synthesized nanoparticles and their stability in BHI medium. Representative TEM image (**A**) and relative size distribution analysis (**B**). DLS spectra of (**C**) 5 nm platinum nanoparticles and (**D**) 5 nm platinum nanoparticles coated with BSA, both dissolved in water (red curves) and BHI medium (blue curves). PtNPs alone are stable and monodispersed in water, but they aggregate when added to BHI medium (**C**). Conversely, PtNPs-BSA show a similar peak when dissolved in water and in BHI medium ((**D**), centered at around 30 nm), confirming the stability ensured by BSA-coating.

**Figure 2 antioxidants-12-01029-f002:**
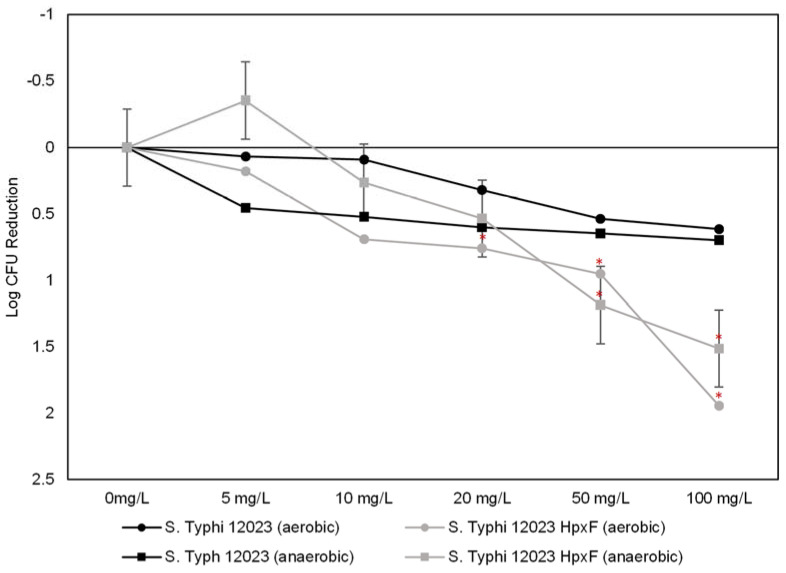
Growth reduction effect of PtNPs on *Salmonella* Typhimurium. Parental strain 12023 (black) showed a weak decrease of the growth. Mutant strain 12023 HpxF^-^ (grey) growth decreased continuously as a function of PtNP dose with a maximum Log CFU reduction of 2 with 100 mg/L of PtNPs.* *p*-value of 0.05.

**Figure 3 antioxidants-12-01029-f003:**
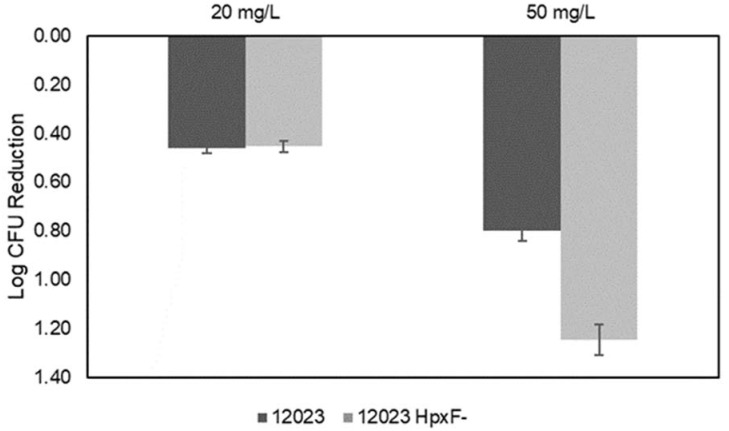
Biocidal effect of sublethal concentrations of PtNPs on *Salmonella* Typhimurium 12023 (black) and 12023 HpxF (grey). The exposure to 20 mg/L PtNPs for 1 h caused a 0.2 and 0.45 Log CFU reduction for wild type and mutant strain, respectively. The 1 h exposure to 50 mg/L showed an increased biocidal effect with 0.46 and 0.8 Log CFU reduction for wild type and mutant strain, respectively.

**Figure 4 antioxidants-12-01029-f004:**
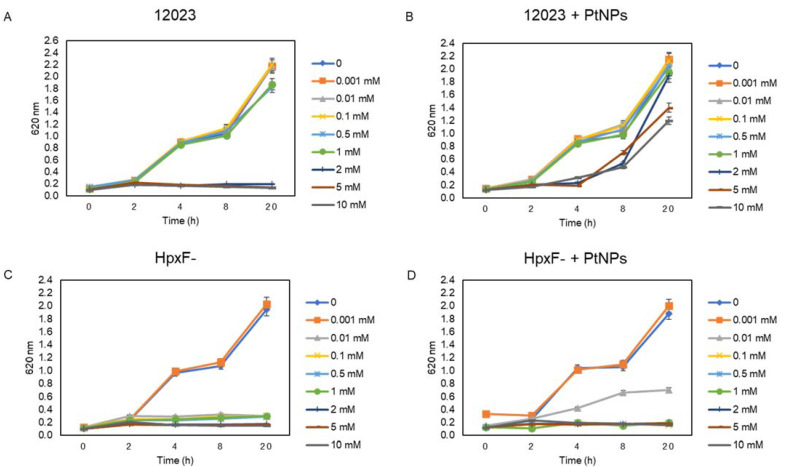
Exposure of *S.* Typhimurium parental strain 12023 to millimolar concentrations of H_2_O_2_ only (**A**) and in combination with 10 µg/mL PtNPs (**B**); and exposure of mutant strain 12023 HpxF^-^ to millimolar concentrations of H_2_O_2_ only (**C**) and in combination with 10 µg/mL PtNPs (**D**).

**Figure 5 antioxidants-12-01029-f005:**
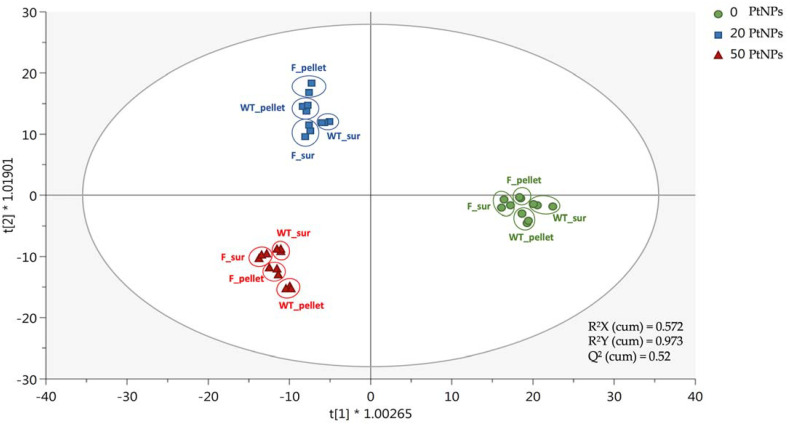
Orthogonal Projections to Latent Structures Discriminant Analysis (OPLS-DA) score plot based on the metabolomic profile of treated and untreated supernatant and cells of parental strain 12023 and mutant strain 12023 HpxF^-^. The exposure was carried out with 0 mg/L (green), 20 mg/L (blue) and 50 mg/L (red) of PtNPs. WT_sur: parental strain supernatant sample; WT_pellet: parental strain cell sample; F_sur: mutant strain supernatant sample; F_pellet: mutant strain cell sample.

**Figure 6 antioxidants-12-01029-f006:**
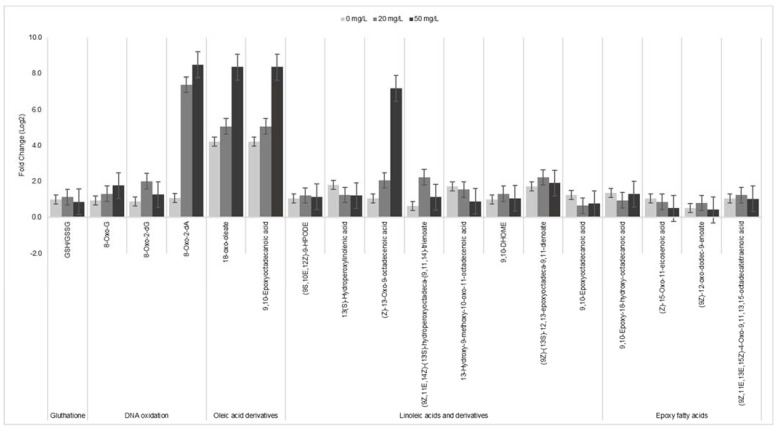
Fold change of the mutant 12023 HpxF^-^ in comparison to parental strain 12023 at a PtNPs dose of 0 mg/L (light grey), 20 mg/L (grey) and 50 mg/L (dark grey). Overall cellular oxidation state was determined as the ratio between reduced (GSH) and oxidized (GSSG) Glutathione. DNA oxidation was determined by markers 8-oxo-G, 8-oxo-2-dG and 8-oxo-2-dA. Lipid peroxidation was assessed with markers for oleic acid, linoleic acid derivatives and epoxy fatty acids.

## Data Availability

Not applicable.

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
