# Peer review of "Role of Platinum Nanozymes in the Oxidative Stress Response of Salmonella Typhimurium"

_antioxidants, 2023, doi:10.3390/antiox12051029_

Round 1

Reviewer 1 Report

In this study, the oxidative stress response of Salmonella enterica serovar Typhimurium cells exposed to 5 nm citrate-coated PtNPs was investigated. It should undergo revision before this work can be published in a scientific journal. 

1.     What is the significance of the role of oxidative stress response in Salmonella Typhimurium, and how do platinum nanozymes interact with this response?

2.     What is the mechanism of action of platinum nanozymes, and how does it relate to the oxidative stress response in Salmonella Typhimurium?

3.     How were the experiments designed and conducted to investigate the role of platinum nanozymes in the oxidative stress response of Salmonella Typhimurium, and what were the key findings?

4.     How do the results of the study contribute to our understanding of the potential applications of platinum nanozymes in combating bacterial infections and diseases?

5.     What are the limitations and challenges of using platinum nanozymes in medical and biological applications, and how can they be addressed in future research?

6.     What are the potential implications of the research for the development of new treatments or therapies for bacterial infections and diseases, and what are the next steps for future research in this area? Compare the work with, Journal of Lightwave Technology 39 (12), 4069-4081, 2021.

7.     What are the potential ethical and safety considerations of using platinum nanozymes in medical and biological applications, and how can they be addressed and regulated?

Author Response

  1. What is the significance of the role of oxidative stress response in Salmonella Typhimurium, and how do platinum nanozymes interact with this response?

The significance of the oxidative stress response in Salmonella Typhimurium is presented in lines 94-102. The mode of action of Pt nanozymes linked to the oxidative stress response of S. Typhimurium is reported in section 3.

  1. What is the mechanism of action of platinum nanozymes, and how does it relate to the oxidative stress response in Salmonella Typhimurium?

The mechanism of action of Pt nanozymes is described in sections 3.2 and 3.3. The use of a mutant strain of S. Typhimurium lacking three catalases and two alkyl-reductases together with the respective parental strain, allowed us to understand that the ROS-coping machinery in Salmonella is strong enough to endure any oxidative stress caused by the PtNPs, no matter the nature of the catalytic activity (oxidase- or peroxidase-like). This was stated in the conclusions.

  1. How were the experiments designed and conducted to investigate the role of platinum nanozymes in the oxidative stress response of Salmonella Typhimurium, and what were the key findings?

The experiments conducted can be revised in section 2. Briefly, A mutant strain of S. Typhimurium ((ΔkatE ΔkatG ΔkatN ΔahpCF ΔtsaA) lacking various of the necessary enzymes to endure oxidative stress, together with its parental strain, were exposed to PtNPs in different conditions. The key findings can be found in section 3. Briefly, the mutant strain was more susceptible to PtNPs, especially under anaerobic conditions. The metabolomics-derived rate of oxidation of glutathione, DNA and membrane-related lipids, confirmed the strong oxidase-like activity of PtNPs. Regarding the parental strain, very limited growth inhibition activity of PtNPs was observed at sublethal levels. The antibacterial activity for both strains was dose-dependent. Additionally, when conducting the experiments in the presence of an exogenous source of ROS, PtNPs showed a switch in catalytic activity similar to that of peroxidases, exerting a protection effect on the parental strain and a mild protection effect on the mutant strain.

  1. How do the results of the study contribute to our understanding of the potential applications of platinum nanozymes in combating bacterial infections and diseases?

The conclusions have been modified to clearly state the contribution of the findings of this study.

  1. What are the limitations and challenges of using platinum nanozymes in medical and biological applications, and how can they be addressed in future research?

The limitations of the use of PtNPs are summarized in the introduction. The conclusions have been modified to clearly state the contribution of the findings of this study and the applications of PtNPs for future research.

  1. What are the potential implications of the research for the development of new treatments or therapies for bacterial infections and diseases, and what are the next steps for future research in this area? Compare the work with Journal of Lightwave Technology 39 (12), 4069-4081, 2021.

The limitations of the use of PtNPs is summarized in the introduction. The conclusions have been modified to clearly state the contribution of the findings of this study and the applications of PtNPs for future research.

  1. What are the potential ethical and safety considerations of using platinum nanozymes in medical and biological applications, and how can they be addressed and regulated?

Safety considerations for food safety are including cytotoxicity are included in the introduction and conclusions. Further considerations are beyond the scope of this study.

Reviewer 2 Report

Role of platinum nanozymes in the oxidative stress response of Salmonella Typhimurium by Daza and others is a significant and focused research article with high readership quality. However manuscript requires further amendment in the presentation and discussions part. Therefore I request authors to revise the manuscript with following points

1) A broad and enhanced introduction is required with incorporation of some more references particularly on silver and gold nanoparticles publications ( PMID;35005942, PMID; 36269783, PMID; 31586898, PMID;30606534). Where authors mentioned the action and reactions of different nakano-articles in cell metabolism.

2) Preparation nad synthesis of PtNP requires more explanation particularly the reaction part.

3) In Fig. 2, Why 5 mg/L of S. Typhi showed the increase and later decreased (Log CFU reduction) . Proper explanation is required for this bifunctional activity.

4) Discussions focused here on S> thyme, although some additional information with respective to other organism adds more taste to manuscript.

5) Did authors used positive control to mimic the enzyme like activity of PtNP.

Author Response

1. A broad and enhanced introduction is required with incorporation of some more references particularly on silver and gold nanoparticles publications (PMID;35005942, PMID; 36269783, PMID; 31586898, PMID;30606534). Where authors mentioned the action and reactions of different nano-articles in cell metabolism.

Thank you for your suggestion. Additional information has been added from lines 48-60 and 69-76.

2. Preparation and synthesis of PtNP requires more explanation particularly the reaction part.

The manuscript has been amended according to the Reviewer’s suggestion (Lines 109-114)

3. In Fig. 2, Why 5 mg/L of Typhimurium showed the increase and later decreased (Log CFU reduction). Proper explanation is required for this bifunctional activity.

Thank you for your revision. In fact, at 5mg/L 12023 HpxF showed an increase in bacterial counts  under anaerobic conditions in comparison to a mild reduction in the remaining strains. The standard deviation for this value was very high indicating high variability among the replicates. Additionally, the values at 5 mg/L for the mutant strain under aerobic and anaerobic conditions were not statistically significant to each other making it pointless to comment these differences. The argument is focused on the fact that with higher PtNPs doses  the mutant is clearly more susceptible than the wild type strain.

4. Discussions focused here on Typhimurium, although some additional information with respective to other organism adds more taste to manuscript.

Thank you for your comment. Additional information was added to the discussion in lines 300-308 and 420-425.

5. Did authors used positive control to mimic the enzyme like activity of PtNP.

One of the aims of this study was to evaluate the different behaviour of PtNPs under aerobic and anaerobic conditions with or without exogenous sources of ROS. The switching nature of PtNPs highlighted in our results prevented us from adding a positive control, as the behaviour was unknown. At this point, no positive control was necessary.

Reviewer 3 Report

 In this manuscript, the authors described “Role of platinum nanozymes in the oxidative stress response of Salmonella Typhimurium”. This paper show that PtNPs exert antimicrobial activity mainly due to their oxidase-like properties. This oxidase-induced effect compromises the integrity of the bacterial membrane and causes DNA damage. Conversely, when combined with other ROS such as H2O2, PtNP peroxidase function promotes ROS scavenging and protects bacterial cells from oxidative damage. However, there are a few points that need to be clarified.

Comment

1.     The resolution of all the pictures is very poor. The author must improve it.

2.     Pt NPs, as nanozymes with tailored catalytic activity and ROS scavenging ability, could play an important role in nanomedicine. Cytotoxicity data for Pt NPs are rather confusing. The author shall be discussion it.

3.     The DCF intensity of Pt NP-treated Salmonella typhi can directly confirm the oxidative stress response. The author shall be adding it.

4.     The SOD activity of Pt NP-treated Salmonella typhi can directly confirm the oxidative stress response. The author shall be adding it.

Author Response

  1. The resolution of all the pictures is very poor. The author must improve it.

Thank you for your comment. The resolution of all figures has been improved.

  1. Pt NPs, as nanozymes with tailored catalytic activity and ROS scavenging ability, could play an important role in nanomedicine. Cytotoxicity data for Pt NPs are rather confusing. The author shall be discussion it.

Thank you for your suggestions. As correctly stated by the Reviewer, PtNPs are promising nanomaterials in the field of nanomedicine, and their key role in the treatment of various pathologies has been widely demonstrated in the past few years. Even though the cytotoxicity of these nanomaterials is still partially under debate, it has been recently reported that it could be mainly related to reaction by-products and release of Pt-ions during the synthesis of PtNPs. Therefore, extensive purification procedures of the prepared batches lead to safe and biocompatible PtNPs, as demonstrated in various scientific publications (among which this review as main example: Chem. Soc. Rev., 2017,46, 4951-4975). Such pure nanomaterials have shown safe applicability for various biomedical applications (ROS scavenging, preventive therapy for cancer and cardiovascular diseases).

  1. The DCF intensity of Pt NP-treated Typhimurium can directly confirm the oxidative stress response. The author shall be adding it.

We thank you for your suggestion. Indirect approaches for the measurement of oxidative stress were adopted including MDA measurement, and metabolomic approach suing DNA, lipid and glutathione oxidation markers. The results provided by these methodologies confirms the oxidative stress response and the differences between the mutant and parental strain.

  1. The SOD activity of Pt NP-treated Typhimurium can directly confirm the oxidative stress response. The author shall be adding it.

We thank you for your suggestion. Indirect approaches for the measurement of oxidative stress were adopted including MDA measurement, and the metabolomic approach suing DNA, lipid and glutathione oxidation markers. The results provided by these methodologies confirms the oxidative stress response and the differences between the mutant and parental strain.

Round 2

Reviewer 1 Report

Satisfactory revision. 

Reviewer 3 Report

accepted